# Diversity Over Frequency: Rethinking Tool Use in Visual Chain-of-Thought Agents

Dong-Hee Kim [1]    Reuben Tan [2]    Donghyun Kim [1]

## Abstract

Visual agents employ external visual tools within visual chains of thought to incorporate fine-grained evidence. While prior work has mainly studied these tools in visual search tasks, their role in more complex visual reasoning remains underexplored. In this paper, we move beyond simple visual search tasks to investigate more challenging tasks, including 3D spatial reasoning and medical visual question answering, where agents must integrate tool-acquired local evidence with the global context. We identify a *tool-use collapse phenomenon*: models progressively stop using tools while still achieving higher task accuracy. Moreover, we observe a clear asymmetry: (i) completely eliminating tool use degrades performance, whereas (ii) incentivizing tool use yields only marginal gains despite substantially increasing usage. We find that vanilla training and tool-use encouragement both reduce rollout diversity, explaining why higher tool use does not yield stronger reasoning performance. Motivated by these findings, we add an entropy regularization term to encourage diverse rollout exploration, achieving the best performance despite gradually declining tool usage. Overall, our findings suggest a training-time view of tools as scaffolding, where broader exploration over language generation and visual tool invocation improves reasoning despite tool-use collapse. Project page: https://scaffolded-exploration.github.io

## 1. Introduction

Recent visual agents increasingly adopt a visual chain-of-thought where external visual tools such as zoom-in cropping, grounding, and other localized visual operations serve as intermediate reasoning steps to capture fine-grained evidence. In benchmarks like V* (Wu & Xie, 2024), employing such tool-mediated reasoning chains has been shown to be critical for solving high-resolution tasks (Zheng et al., 2026; Zhang et al., 2025a; Lai et al., 2026). This formulation frames the iterative execution of tools as the reasoning process itself, a strategy that is further exemplified by practical systems like ChatGPT-o3 (OpenAI, 2025). Consequently, there is growing interest in training agents to actively leverage these tool-based reasoning paths for broader visual perception challenges (Shao et al., 2024; Lai et al., 2026; Zheng et al., 2026; Su et al., 2026; Wu et al., 2025; Su et al., 2025; Zhang et al., 2025a).

However, it remains unclear whether such tool-driven gains generalize to broader, more diverse, and more complex visual tasks. Most prior studies focus on simple visual search scenarios where merely zooming into or cropping relevant regions is sufficient to solve the task. In contrast, we investigate visual tasks that demand more advanced reasoning, using 3D spatial reasoning from 2D images as our main testbed for general spatial understanding and further validate our findings on medical visual question answering (VQA). Unlike visual search tasks that can be solved through simple localization or detection, spatial reasoning requires identifying implicitly relevant regions and understanding their relationships within the global spatial structure. Thus, this task requires agents to learn selective tool use, reasoning about when additional visual evidence is beneficial for complex spatial understanding. Studying this regime is crucial because, as we show, naive application of reinforcement learning (RL) struggles to regulate selective tool use in complex spatial reasoning tasks, leading to unexpected failure modes.

Our first finding is a consistent training pathology we call tool-use collapse. Starting from a pretrained tool-using agent, vanilla reinforcement learning fine-tuning (RFT) improves accuracy while driving the tool-use rate, defined as the fraction of rollouts that invoke a tool, to near zero. This collapse occurs even when employing methods designed to mitigate length penalties associated with multi-step tool-based reasoning (Lai et al., 2026). In contrast, tool-encouraging reward design (Zheng et al., 2026; Su

---

[1]Department of Artificial Intelligence, Korea University, Seoul, South Korea [2]Microsoft Research, Redmond, USA. Correspondence to: Donghyun Kim <d_kim@korea.ac.kr>.

*Proceedings of the 43rd International Conference on Machine Learning*, Seoul, South Korea. PMLR 306, 2026. Copyright 2026 by the author(s).

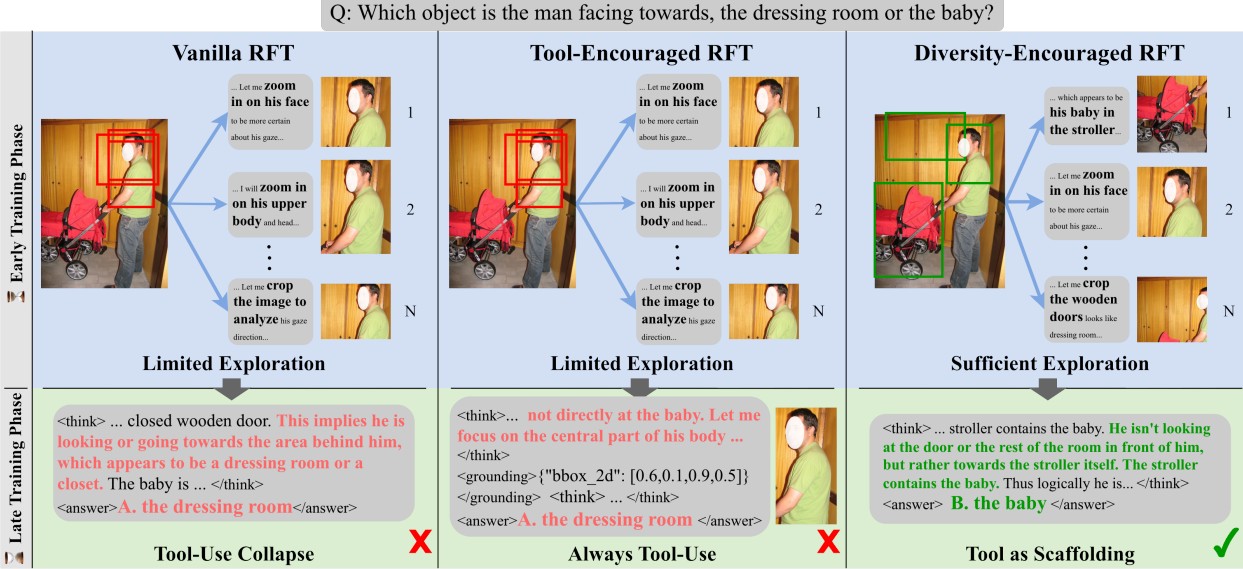

Figure 1. **Effect of Early Exploration on the Late Training Phase.** This figure compares agent behaviors in the early training (top) and late training (bottom) phases. Both vanilla (Left) and tool-encouraged RFT (Middle) suffer from limited early exploration in both text and visual modalities. This lack of rollout diversity restricts models to local optima with marginal gains regardless of tool usage frequency. In contrast, encouraging exploration (Right) leads to better performance even when the agent rarely uses tools. We call this the scaffolding effect where early tool use acts like temporary supports to build reasoning capabilities that remain effective even after tool use is discarded.

et al., 2026) in RFT consistently uses tools but yields only marginal accuracy gains. Fig. 1 illustrates this asymmetry: vanilla RFT (left) and tool-encouraging RFT (middle) show that both fine-tuning approaches suffer from limited early exploration, repetitively fixating on salient features (e.g., the man's face). Conversely, the right column demonstrates that fostering diverse exploration acts as *scaffolding* by covering broader contexts (e.g., the stroller and wooden door), enabling the accurate inference. To make this mechanism explicit, we quantify exploration in both text and vision, which reveals that vanilla RFT and tool-encouraged RFT lead to diversity degradation in both modalities. This confirms that the scaffolding effect relies not on tool frequency, but on preventing premature convergence to repetitive reasoning and visual fixation.

Motivated by these findings, we employ adaptive entropy regularization (Zhang et al., 2025b) during RFT to encourage such diverse exploration, resulting in the best performance despite tool usage gradually declining during training. Overall, our results support a training-time view of tools as scaffolding, where early tool-mediated exploration shapes representations that enable accurate tool-less inference.

In summary, our contributions are as follows:

1. We identify tool-use collapse under vanilla RFT of a pretrained tool-using visual agent for visual reasoning tasks (e.g., 3D spatial reasoning), where accuracy improves while tool use almost vanishes.

2. We show a clear asymmetry: removing tools degrades performance, whereas reward-based tool encouragement while RFT drives constant tool calls with only marginal gains. We link both outcomes to insufficient exploration measured in both textual reasoning and visual tool behavior.

3. We show that encouraging exploration by adding an adaptive entropy regularization term increases rollout diversity during training and achieves the best performance while tool usage still naturally declines, supporting the view of tools as scaffolding during training. We further provide an additional discussion study on VQA-RAD to examine whether these dynamics extend to medical visual question answering.

## 2. Preliminaries & Experimental Setup

### 2.1. Agentic Visual Tool Interface

We consider an visual agents that solves visual reasoning tasks through a structured interaction protocol. Given an input image and a text query, the model may either answer directly from the initial visual observation or invoke external visual tools through special trigger tokens. These tools can include operations such as region grounding, object detection, segmentation, point selection, or image cropping, depending on the available interface.

**Agentic Interaction Loop.** Given a spatial reasoning text query $q$ and input image $I_0$, the policy model $\pi_\theta$ generates a rollout $\tau$ through a thought–action–observation loop.

A rollout $\tau$ comprises a single text-only step or a multi-turn sequence of generated `<think>` spans, optional tool-triggered actions, and resulting observations until the model outputs a terminal `<answer>`.

At step $i$, the model produces:

- **Thought** $T_i$: Internal reasoning conditioned on the interaction history $h_{<i} = \{(T_j, A_j, O_j)\}_{j<i}$ and current observation.

- **Action** $A_i$: Either emit a tool trigger token with tool-specific parameters, continue reasoning, or terminate with a final `<answer>`.

- **Observation** $O_i$: The resulting image patch or terminal state.

The rollout terminates when the model outputs a final answer or interaction turns. In practice, rollouts follow explicit tags, including an initial `<think>` span, an optional tool trigger span with parameters, and a terminal `<answer>` span.

A tool-triggered action is parameterized by a tool type and a set of tool-specific arguments:

$$A_i^{\texttt{tool}} = (\texttt{type}, \texttt{args}), \tag{1}$$

where `type` specifies the visual operation to execute, and `args` contains the corresponding parameters. For example, different tools may take bounding boxes, object queries, masks, or point coordinates as arguments.

**Observation Generation.** Executing a tool action produces observation $O_i$ according to the selected tool and its arguments. The resulting observation is appended to the conversation history, enabling the model to reason over both the initial global observation from $I_0$ and tool-acquired evidence from $\{O_j\}$.

**Tool-Based vs. Tool-Free Rollouts.** A rollout $\tau$ may invoke one or more visual tools multiple times (tool-based) or proceed directly to `<answer>` using only the initial global observation $I_0$ (tool-free). We quantify tool utilization using the Tool Use Ratio, defined as the fraction of rollouts containing at least one tool trigger token. Tool-based rollouts are typically longer in token count and interaction turns, as each tool invocation introduces additional observations, visual tokens, and reasoning steps. This asymmetry becomes critical during RFT, where optimization pressure can favor shorter, lower-variance paths, a phenomenon we analyze in Sec. 3.1.

## 2.2. Reinforcement Learning Objective

We optimize the policy $\pi_\theta$ via reinforcement learning to maximize the expected reward on spatial reasoning tasks:

$$\mathcal{J}(\theta) = \mathbb{E}_{q \sim \mathcal{D}, \, \tau \sim \pi_\theta(\cdot|q,I_0)} \big[ R(\tau, q) \big], \tag{2}$$

where $q$ denotes a question sampled from the training distribution $\mathcal{D}$, $\tau$ is a sampled agentic rollout that may be tool-free or tool-based, and $R(\tau, q)$ evaluates the final correctness of the terminal response in $\tau$ for $q$.

We optimize the policy using a group-based reinforcement learning objective (Guo et al., 2025). For each question, multiple rollouts are sampled from the current policy, assigned outcome rewards, and compared within the group to estimate relative advantages. This formulation is suitable for agentic tool-use settings because rollouts may differ not only in their final answers, but also in whether, when, and how they invoke external visual tools.

**Policy Optimization.** For each question $q$, we sample a group of $G$ rollouts $\{\tau_i\}_{i=1}^G$ from the current policy $\pi_{\theta_{\text{old}}}(\cdot \mid q, I_0)$ and compute rewards. The policy is updated via clipped importance sampling with group-normalized advantages:

$$\mathcal{J}_{\text{GRPO}}(\theta) = \mathbb{E} \left[ \frac{1}{G} \sum_{i=1}^{G} \min\Big( \rho_i A_i, \; \text{clip}(\rho_i, 1 - \epsilon, 1 + \epsilon) \, A_i \Big) \right]. \tag{3}$$

where $\rho_i = \pi_\theta(\tau_i \mid q, I_0) / \pi_{\theta_{\text{old}}}(\tau_i \mid q, I_0)$ and $A_i = (r_i - \mu_r)/\sigma_r$ with group statistics $\mu_r, \sigma_r$.

A practical issue in agentic tool-use learning is that tool-based rollouts are often longer than tool-free rollouts. If over-budget rollouts are treated as failures, the policy can be implicitly biased against tool invocation. We therefore distinguish final-answer failures from budget-exceeding rollouts, and exclude the latter from advantage computation rather than assigning them negative rewards.

## 3. Collapse, Explosion and Exploration Control

**Experimental Instantiation.** While the preceding interface is defined for general agentic visual tool use, our experiments instantiate it using Mini-o3 (Lai et al., 2026). Mini-o3 is a `Qwen2.5-VL-7B-Instruct` (Bai et al., 2025)-based VLM trained with supervised fine-tuning (SFT) and RFT to perform iterative visual tool use through a structured interaction format. We initialize all experiments from the pretrained Mini-o3 weights.

In this instantiation, the available visual tool is a zoom-in grounding operation. The model emits a `<grounding>` trigger span with parameters:

$$A_i^{\texttt{<grounding>}} = (\texttt{bbox\_2d}, \texttt{source}), \tag{4}$$

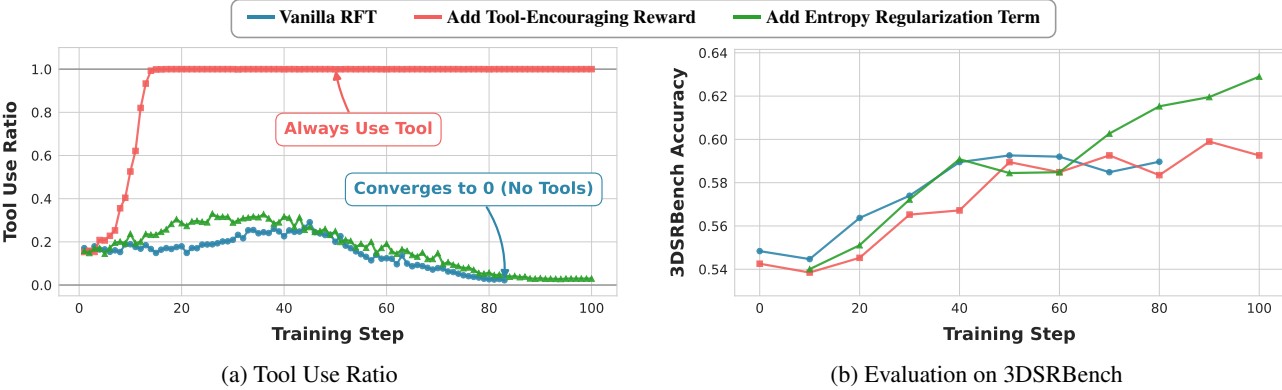

(a) Tool Use Ratio            (b) Evaluation on 3DSRBench

*Figure 2.* **Analysis of tool usage ratio and performance across training steps.** (a) Tool Use Ratio over training defined as the fraction of rollouts that contain at least one grounding action. (b) Validation accuracy trends. *Vanilla RFT* (blue) and *Tool-Encourage reward* (red) exhibit divergent tool usage behaviors despite similar limited accuracy gains. In contrast, adding an *entropy regularization term* (green) encourages exploration which leads to a significant increase in final performance.

where `bbox_2d` $\in [0, 1]^4$ specifies normalized coordinates $(x_1, y_1, x_2, y_2)$, and `source` $\in \{I_0, O_1, \ldots, O_{i-1}\}$ indicates whether the crop is taken from the original image or from a previous observation. Executing this action produces a cropped image patch that is appended to the interaction history. This concrete instantiation allows us to study a broader question: how reinforcement learning reshapes the tool-trigger behavior of agentic VLMs. Although the tool in our experiments is zoom-in grounding, the same training dynamics can arise whenever a model must decide whether to invoke an external visual operation, how often to invoke it, and how much additional observation context to incorporate.

**RL Implementation.** In our experiments, we follow Mini-o3 (Lai et al., 2026) and adopt DAPO (Yu et al., 2025), a stabilized group-based policy optimization method built on GRPO (Guo et al., 2025). We use the same stabilization techniques as Mini-o3, including clip-higher, dynamic sampling, token-level policy loss, and over-turn masking. In this concrete setting, over-turn masking excludes rollouts that exceed the maximum interaction budget from advantage computation rather than treating them as failures. This prevents the policy from being implicitly penalized for invoking the `<grounding>` tool.

In practice, we report results up to 100 training steps. Beyond this point, we observe oscillatory dynamics where validation performance no longer improves reliably. We additionally note a practical limitation induced by DAPO's dynamic sampling. When a sampled group yields degenerate rollouts, for example when all rewards are identical, the batch is rejected and rollouts are regenerated from a new batch. If the number of such resampling attempts exceeds 10 within a single update step, the wall-clock cost increases sharply, making continued training inefficient. We therefore terminate runs at this point.

**Training Data.** We train on 1.2k 3D spatial reasoning samples derived from SpatialReasoner (Ma et al., 2025b), formatted as (image, question, answer) triplets. Questions require understanding of 3D spatial relationships such as relative height, depth ordering, and orientation that cannot be reliably inferred from 2D image coordinates alone. Images are sourced from OpenImages (Kuznetsova et al., 2020) and depict diverse real-world scenes with multiple objects. Critically, questions often reference objects that are small or ambiguous in the global view, creating opportunities where tool-based zoom-in actions could aid spatial reasoning.

**Evaluation Benchmarks: 3DSRBench.** We select 3DSR-Bench (Ma et al., 2025a) as our primary testbed because it presents a mixed-utility environment essential for studying selective tool use. Unlike visual search tasks dominated by small targets, 3DSRBench covers a broad spectrum of object scales (see Appendix C.2 for a detailed comparison), requiring agents to distinguish when tools are truly necessary. The benchmark contains 5,250 multiple-choice questions on MS-COCO images across four types: Height, Location, Orientation, and Multi-object. Questions avoid 2D shortcuts (e.g., camera pitch variations decorrelate vertical position in image vs. 3D height). Model outputs can be free-form (e.g., `C`, `the answer is C`, or `C. ...`); we therefore compute accuracy using a fixed VLM-as-judge (`Qwen2.5-VL-7B-Instruct`) that normalizes each response and determines which option it supports, then compares it against the ground-truth option. All evaluation is reported as Avg@8, where we sample 8 independent generations per question and average the resulting accuracy.

### 3.1. Vanilla RFT Leads to Tool-Use Collapse

We first investigate the behavior of visual agents under vanilla RFT conditions to establish a baseline and analyze the necessity of visual tools.

**Examine Transferable Agentic Ability.** We train the Mini-o3 agent using standard GRPO with DAPO stabilization, as described in Section 2.1, on 1.2k spatial reasoning questions. The model is warm-started from a SFT-RFT checkpoint that is already trained to use `<grounding>` actions.

**The Phenomenon of Tool-Use Collapse: Accuracy Rises as Tool-Use Vanishes** As shown in Figure 2, we observe a distinct training dynamic termed **tool-use collapse**. While the validation accuracy on 3DSRBench improves steadily to reach 59.2% at saturation, the tool use ratio declines monotonically, converging to near 2% by 80 training steps. Notably, the agent maintains a non-trivial tool usage ratio (around ∼20%) during the early stage of training before the collapse accelerates.

**Optimization Preference for No Tools.** We hypothesize that this collapse is driven by an asymmetry in the optimization landscape. A tool-based rollout requires a multi-step interaction loop that generates a `<grounding>` action, processes additional visual tokens from the cropped image, and integrates multi-turn observations. In contrast, tool-free reasoning, producing a direct answer from the global image observation without invoking any tool, follows a shorter rollout. Crucially, this trend still appears even when we attempt to mitigate GRPO's implicit preference for shorter generations by replacing the length-sensitive term with DAPO's token-level policy loss (Yu et al., 2025). While this does not by itself establish a causal mechanism, it suggests that tool-use collapse cannot be explained solely as an artifact of GRPO's length bias.

Finally, to rule out the concern that tool-use collapse is an artifact of training on a small 1.2k dataset, we repeat the same training protocol on the larger SAT dataset (6k samples) (Ray et al., 2025) and observe a similar collapse. Detailed results on SAT training are provided in Appendix B.1.

### 3.2. Ablation: Are Tools Useful?

The collapse observed in the previous experiment raises a critical question: If the agent discards tools and still achieves high accuracy, are tools genuinely unnecessary? To verify this, we conduct an ablation study where tools are completely prohibited.

**Tool-Banned Training and Evaluation.** We train an otherwise identical agent but enforce a strict ban on tool usage throughout both training and evaluation. Specifically, we prevent the model from emitting the tool trigger tags (`<grounding>`, `</grounding>`) during rollout generation and do not execute any zoom-in/crop operations. As a result, the agent must rely exclusively on tool-free reasoning from the initial global observation, without receiving any tool-mediated visual inputs at any stage.

**Early Tool Experience Matters.** The tool-banned agent achieves a final accuracy of 58.1% which represents a 1.1% drop compared to vanilla RFT. This reveals a paradox where the vanilla RFT outperforms the strictly tool-free agent despite eventually abandoning tool usage itself.

**Tools as Training-Time Scaffolding.** This result suggests that the early tool experience (approximately ∼20% tool-use ratio in the baseline) was not wasted but served as training-time scaffolding. Even though tool usage later fades, these early tool-augmented rollouts may provide a richer exploration history and learning signals that transfer to the eventual tool-free policy. In contrast, the tool-banned training, lacking this initial tool-mediated exploration, fails to reach the same performance peak.

### 3.3. Do Incentive Reward Designs Improve Tool Use and Accuracy?

Sec. 3.1 and Sec. 3.2 revealed a paradoxical pattern. Under standard RL fine-tuning, tool usage collapses toward zero while accuracy improves. Yet removing tools entirely from the beginning reduces the final accuracy (58.1% vs. 59.2%). This indicates that tool interactions can benefit learning dynamics even if tools are rarely used at convergence. We therefore ask a sharper question: If early tool exposure matters, should we deliberately increase tool usage to improve performance?

A key premise of our analysis is that tool usage frequency is not the objective.

**Tool-Encouraging Reward Design.** We compare two representative tool incentivization schemes. While Mini-o3 (Lai et al., 2026) encouraged tool usage passively by omitting negative rewards for long multi-turns, we introduce an explicit reward to directly incentivize tool use.

**(1) Tool bonus.** Following DeepEyes (Zheng et al., 2026), we use a rollout-level reward that combines final correctness with a conditional bonus for successful tool use:

$$R_{\text{DE}}(\tau, q) = \mathbb{I}[y(\tau) = y^*(q)] \\ + \lambda_{\text{tool}} \, \mathbb{I}[y(\tau) = y^*(q)] \, \mathbb{I}[u(\tau) = 1] \,, \quad (5)$$

where $y(\tau)$ is the extracted option label using the fixed VLM-as-judge protocol described in Sec. 2.1, $y^*(q)$ is the ground-truth label, and $u(\tau)$ indicates whether the rollout contains at least one `<grounding>` action.

**(2) Curiosity reward.** Following PixelReasoner (Su et al., 2026), we augment correctness with a curiosity bonus that encourages tool-mediated reasoning when the policy under-

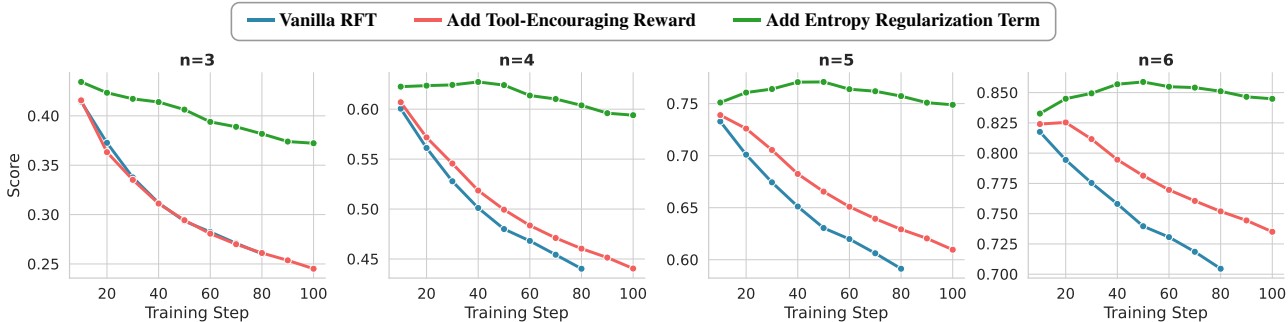

*Figure 3.* **Analysis of Textual Exploration During Training.** We report the distinct $n$-gram ratio (Li et al., 2016) ($n \in \{3, 4, 5, 6\}$) to measure token diversity of generated rollouts. Both the *Vanilla RFT* (Blue) and *Tool-Encourage* (Red) suffer from diversity degradation, indicating increasingly repetitive reasoning patterns. In contrast, adding an *Entropy regularization term* (Green) maintains higher textual diversity, suggesting that the scaffolding effect relies on preventing premature convergence in the reasoning space.

adopts it for a given query:

$$
\begin{aligned}
R_{\text{PR}}(\tau, q) = \; & \mathbb{I}[y(\tau) = y^*(q)] \\
& + \alpha \, \max\big(H - \text{RaPR}(q), 0\big) \, \mathbb{I}[u(\tau) = 1] \\
& + \beta \, r_{\text{penalty}}(\tau),
\end{aligned}
\tag{6}
$$

where $\text{RaPR}(q)$ denotes the rate of tool-mediated reasoning for query $q$, following the Rate of Pixel-space Reasoning definition in PixelReasoner. Concretely, we estimate $\text{RaPR}(q) = \mathbb{E}_{\tau \sim \pi_\theta(\cdot|q, I_0)}[u(\tau)]$, which corresponds to the expected frequency of `<grounding>` usage for query $q$, and $H$ is the target adoption threshold. Following Pixel-Reasoner, we impose an upper bound on tool calls within a rollout via $r_{\text{penalty}}(\tau) = \min\big(N - n_{\text{tool}}(\tau), 0\big)$, where $n_{\text{tool}}(\tau)$ is the number of tool calls in $\tau$. This penalty is 0 when $n_{\text{tool}}(\tau) \leq N$ and becomes negative only when tool calls exceed $N$.

For both schemes, we follow the original papers' hyperparameter choices (e.g., $\lambda_{\text{tool}}$, $\alpha$, $\beta$, $H$, and $N$) in our experiments. For clarity, we report the Tool bonus results in the main text. The Curiosity reward shows the same trend and is deferred to Appendix B.2.

**Tool Explosion.** The results are shown in Fig. 2. Under the DeepEyes' tool bonus, the policy rapidly increases its reliance on tool calls. Starting from the same warm-started checkpoint (Tool Use Ratio ~20%), the ratio rises throughout training and reaches 100% at saturation, meaning that every rollout invokes at least one tool call. Despite this persistent tool usage, the final accuracy on 3DSRBench increases only modestly to 59.9% (vs. 59.2% for the baseline).

**Increased tool usage does not necessarily correspond to greater diverse exploration.** Taken together with the tool-banned ablation in Sec. 3.2, these results indicate that the limitation is not the mere availability of tools. Tools can provide a useful learning signal early in training, yet simply driving tool usage to 100% yields only marginal gains. This motivates a closer diagnosis of what changes in learning

| Method | mIoU (↓) | CLIP (↑) | Visual Behavior |
|---|---|---|---|
| Vanilla RFT | 0.554 | 0.184 | High Fixation |
| Tool-Encourage | 0.557 | 0.187 | High Fixation |
| Entropy-Regularized | 0.494 | 0.184 | Active Exploration |

*Table 1.* **Quantitative Analysis of Visual Exploration.** We measure spatial diversity via mIoU computed between crop boxes across rollouts for a single query (lower indicates broader exploration) and semantic relevance via CLIP alignment between the crop and a keyword prompt. We form the keyword prompt from question nouns (the red words in Fig. 4). Higher CLIP indicates better alignment; absolute values are reported for comparison. *Entropy-Regularized* promotes active visual exploration without losing semantic focus, whereas others suffer from spatial fixation.

dynamics under vanilla RFT and Tool-Encourage RFT.

Specifically, we ask whether these training runs maintain diverse reasoning rollouts as learning progresses. We characterize diversity along two axes that correspond to our tool interface: (i) the textual reasoning diversity that precedes a crop decision, and (ii) the visual regions diversity selected by crop actions.

To quantify diversity in the textual reasoning space, we analyze the reasoning produced before issuing a crop via `<grounding>`. This choice is motivated by an entropy probe: we found that uncertainty is not primarily expressed in the `<grounding>` box coordinates themselves, but earlier in the preceding textual reasoning, suggesting that the pre-`<grounding>` span often drives where the agent decides to crop. Following the interaction format in Sec. 2.1, each rollout contains an initial `<think>...</think>` span that precedes either a `<grounding>` action (tool-based) or a direct answer (tool-free), enabling consistent extraction of the pre-`<grounding>` reasoning span across methods. As shown in Fig. 3, both vanilla RFT (red) and Tool-Encourage (blue) exhibit a monotonic decrease in $n$-gram diversity over training, indicating increasingly templated pre-`<grounding>` reasoning.

We also diagnose the visual side of exploration, since higher tool usage does not necessarily imply broader visual evidence. Our goal is to verify, in order, (ii-a) whether the

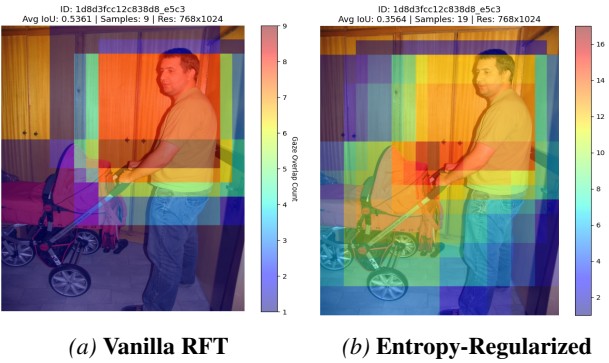

*(a)* **Vanilla RFT**    *(b)* **Entropy-Regularized**

> **Q:** *Which object is the* *man* *facing towards,* *the dressing room* *or* *the baby*?

*Figure 4.* **Visual Exploration During Training.** Comparison of crop distributions at an early-training checkpoint where tool use still actively explores. While the baseline **(a)** narrowly focuses on the salient subject, the entropy-regularized model **(b)** actively explores contextually relevant regions (the stroller) referenced in the query, demonstrating wider exploration.

model explores diverse crop locations in image space, and (ii-b) whether the visited crops remain semantically tied to the question rather than drifting to irrelevant regions.

Concretely, we take the checkpoint at step 40, where vanilla RFT training exhibits the most active tool use, and compare it against the Tool-Encourage checkpoint at the same step. For each image–question pair in the 1.2k training set, we sample 50 stochastic rollouts from each policy and collect all crop boxes produced via `<grounding>`. To quantify diversity of crop locations, we compute the mean pairwise IoU between crop boxes generated for the same image–question pair; lower IoU indicates that the policy visits a broader set of regions, whereas higher IoU indicates visual fixation. To quantify semantic relevance, we compute a CLIP score between each cropped region and a keyword extracted from the question (nouns; e.g., the red-highlighted words in Fig. 4), and average scores across rollouts.

Tab. 1 summarizes the results. Tool-Encourage invokes `<grounding>` substantially more often than vanilla training (roughly three times more crops per query), but the crop-location statistics remain similar across the two methods (mean pairwise IoU $> 0.55$ in both cases). For example, Fig. 4a shows that vanilla RFT concentrates its crops around the man's upper body. Moreover, the semantic relevance of selected crops is comparable (similar CLIP scores), indicating that increased tool use does not translate into broader coverage or more question-aligned visual evidence. These results reinforce that simply encouraging tool invocation is insufficient to expand the exploration history during training, motivating the need to explicitly sustain exploration during RL fine-tuning in the next section.

## 3.4. Does Diversity Matter? Encouraging Exploration via Entropy Regularization

The results in Sec. 3.3 show that increasing tool usage via reward incentives is insufficient to sustain exploration. We therefore treat tools as a means to broaden exploration history during training and use entropy regularization as a simple control knob to prevent premature convergence.

**Entropy Regularization as an Exploration Control Knob.** We augment the GRPO objective with a policy-entropy regularization term:

$$\mathcal{J}_{\text{ent}}(\theta) = \mathcal{J}_{\text{GRPO}}(\theta) + \lambda_t \cdot \mathbb{E}_{q,\tau}\left[\bar{\mathcal{H}}(\tau)\right], \qquad (7)$$

where $\lambda_t \geq 0$ is the entropy coefficient at update step $t$ and $\bar{\mathcal{H}}(\tau)$ is the mean token-level entropy over the entire rollout $\tau$. For an autoregressive policy, token-level entropy at generation step $k$ with state $s_k$ is

$$\mathcal{H}(\pi_\theta(\cdot \mid s_k)) = -\sum_{v \in \mathcal{V}} \pi_\theta(v \mid s_k) \log \pi_\theta(v \mid s_k), \quad (8)$$

and we compute $\bar{\mathcal{H}}(\tau)$ by averaging $\mathcal{H}(\pi_\theta(\cdot \mid s_k))$ across all generated tokens in the rollout.

In preliminary experiments, we found a fixed entropy coefficient to be brittle. When it is too small, it has negligible effect on exploration, whereas larger values can destabilize training and lead to degenerate generations, such as mixed-language outputs or repetitive token loops. To avoid manual tuning, we adopt an adaptive entropy coefficient using proportional feedback, following prior work (Zhang et al., 2025b):

$$\lambda_t = K_p \left[\mathcal{H}_{\text{target}} - \mathcal{H}_t\right]_+, \qquad (9)$$

where $\mathcal{H}_t$ is the current batch entropy measured over full rollouts, $\mathcal{H}_{\text{target}} = 0.9$ is the target entropy, $K_p = 0.03$ is the gain, and $[\cdot]_+ = \max(\cdot, 0)$. This rule increases exploration pressure only when entropy drops below the target and otherwise sets $\lambda_t$ to zero.

**Entropy Regularization Improves Best Accuracy.** With entropy regularization, the agent reaches a best validation accuracy of 62.9% on 3DSRBench during training. Although the run starts from the same warm-started checkpoint (Tool Use Ratio ∼20%), the Tool Use Ratio decreases over training and is 3% at the end of training (measured on training-batch rollouts). We report this run as the green curve in Fig. 2 for consistency with Fig. 3.

**Explore More, Tools Fade, Performance Persists.** This training also uses tools only rarely by the end of training, yet it substantially outperforms both the vanilla RFT and the reward-incentivized setting. Notably, directly incentivizing

| Method | Tools? | Acc. | Tool Usage (Init→Sat) |
|---|---|---|---|
| vanilla RFT | Yes | 59.2% | ∼20% → ≈2% |
| Tool-banned | No | 58.1% | 0% → 0% |
| Reward-encouraging reward design | Yes | 59.9% | ∼20% → 100% |
| Entropy-regularized | Yes | **62.9%** | ∼20% → ≈3% |
| Tool-banned & Entropy-regularized | No | 57.8% | 0% → 0% |

*Table 2.* **Experiment Summary.** Vanilla RFT collapses tool use (20%→2%), banning tools hurts accuracy, reward incentives saturate tool use (20%→100%) with marginal gains, and entropy regularization yields the best accuracy with low tool use at saturation.

tool invocation drives near-constant `<grounding>` calls but yields only marginal gains, which makes it difficult to attribute the improvement to learning to use tools more at inference time. Instead, the evidence points to a training-time effect: what matters is not tool frequency itself, but whether training sustains exploration in the reasoning process and the visual evidence it visits.

This interpretation is supported by our exploration diagnostics in both text and vision. In the textual reasoning space, Fig. 3 green line shows that the entropy-regularized run maintains higher $n$-gram diversity, whereas both vanilla RFT and Tool-Encourage become increasingly templated in the pre-`<grounding>` `<think>` span. In the visual tool space, Tab. 1 shows that the entropy-regularized policy explores more diverse crop locations (mIoU 0.494 vs. 0.554/0.557 for Vanilla/Tool-Encourage) while preserving comparable semantic relevance (CLIP 0.184 vs. 0.184/0.187), and Fig. 4b provides a qualitative example of this broader coverage. Taken together, these results suggest that the key benefit comes from a richer exploration history during training. As in vanilla RFT, tool usage still fades late in training, but the resulting tool-free policy is stronger because it was shaped by more diverse reasoning rollouts and visual evidence earlier on.

At the same time, this result raises an important follow-up question. Is the improvement driven by a general effect of maintaining higher policy entropy, or does it specifically depend on tool-mediated visual exploration during training? To disentangle these factors, we next compare entropy regularization with and without tool access. We provide additional transfer experiments and a direct early-phase ablation supporting this interpretation in App. B.

### 3.5. Ablation: Is the Entropy Gain Tool-Dependent?

**Tool-Banned Entropy Regularization.** To test whether entropy regularization yields tool-agnostic gains, we repeat the tool-banned protocol in Sec. 3.2 while adding the same entropy control as in Section 3.4.

**Entropy Gains Require Tool Access.** Under strict tool banning, entropy regularization does not improve performance. The tool-banned + entropy run reaches a best validation accuracy of 57.8%, which is not higher than the tool-banned baseline and remains far below the tool-enabled

| Method | 3DSRBench (Acc, %) | CV-Bench-3D (Acc, %) |
|---|---|---|
| *Generalist Baseline* | | |
| Qwen2.5-VL 7B (Bai et al., 2025)[*] | 48.4 | **82.9** |
| *Visual Agents (Zero-shot)* | | |
| DeepEyes (Zheng et al., 2026) | 51.6 | 76.7 |
| Mini-o3 (Lai et al., 2026) | **54.5** | 77.6 |
| *Specialist Reference* | | |
| SpatialReasoner (Ma et al., 2025b)[*] | 60.3 | 80.3 |
| *Ours Experiments (RL Fine-tuning)* | | |
| vanilla RFT (Base) | 59.2 | 76.7 |
| w/ Tool-Encourage Reward | 59.9 | 74.5 |
| w/ Entropy-Regularization | **62.9** | 78.8 |

*Table 3.* **Impact of Exploration on Generalization.** We compare performance on 3DSRBench and CV-Bench-3D (Tong et al., 2024). While vanilla RFT and tool-encourage rewards degrade performance on the general benchmark compared to the pre-trained Mini-o3 weight, Entropy-Regularization is the only method that improves both specialized and general capabilities, confirming that diverse exploration shapes more robust representations. (* are from SpatialReasoner (Ma et al., 2025b))

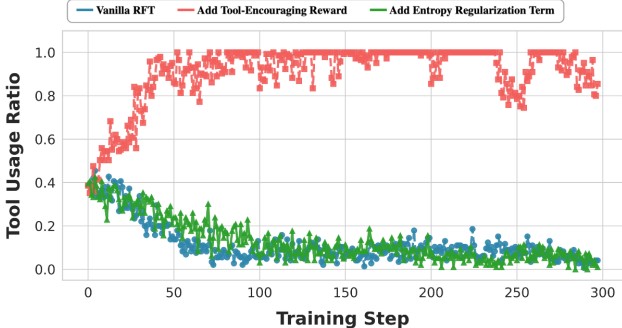

*Figure 5.* Tool use ratio over training for OpenThinkIMG (Su et al., 2025) on VQA-RAD (Lau et al., 2018), defined as the fraction of rollouts that invoke at least one tool.

entropy-regularized setting (62.9%). This contrast indicates that the gains from entropy regularization are not purely tool-agnostic, but depend on tool-mediated visual exploration during training. When tools are unavailable, increasing exploration pressure can be neutral or even detrimental, suggesting that the scaffolding benefit arises from exploring additional visual evidence rather than from entropy alone.

## 4. Discussion

### 4.1. Generalization on Broader Tools and Tasks

To examine whether our findings extend beyond the main experiments, we conduct an additional generalization study using OpenThinkIMG (Su et al., 2025) on VQA-RAD (Lau et al., 2018). Unlike our main experiments, which instantiate tool use with a crop-based zoom-in tool, OpenThinkIMG provides a heterogeneous visual tool suite including OCR, object detection, segmentation, point localization, and axis-drawing tools. This allows us to test whether the same tool-use dynamics appear in a different model configuration, task domain, and tool interface.

Figure 5 shows qualitatively similar training dynamics.

| Method | Accuracy |
|---|---|
| Vanilla RFT | 46.34 |
| Tool-Encouraged RFT | 47.23 |
| Entropy Regularization | 48.78 |

*Table 4.* **Final validation accuracy on VQA-RAD.**

Vanilla RFT and Entropy Regularization both reduce tool use throughout training, while Tool-Encouraged RFT drives the policy to rely on tools almost always. This mirrors the core asymmetry observed in our main experiments, where simply increasing tool frequency does not necessarily translate into stronger task performance.

The final performance ordering also remains consistent, as shown in Table 4. Vanilla RFT achieves 46.34% accuracy, Tool-Encouraged RFT improves modestly to 47.23%, and Entropy Regularization achieves the best result at 48.78%. These results suggest that the central phenomenon studied in this paper is not specific to 3D spatial reasoning, Mini-o3, or crop-based zoom-in perception. Instead, they provide additional evidence for our training-time view of tools as scaffolding, where diverse exploration over language generation and visual tool invocation can improve final reasoning performance even as tool use naturally declines.

### 4.2. Generalization Analysis on CV-Bench-3D

To verify whether adding the entropy-regularization term yields genuine improvements on spatial understanding rather than merely overfitting to the target task, we extended our evaluation to CV-Bench-3D (Tong et al., 2024). This benchmark consists of 1,200 questions that evaluate fundamental depth perception capabilities, including depth order and relative distance estimation from monocular images.

As presented in Tab. 3, Vanilla RFT exhibits a notable trade-off between task-specific optimization and general visual understanding. Comparing against the pre-trained Mini-o3 accuracy of 77.6%, the standard vanilla RFT results in a slight regression to 76.7%. The performance degradation is even more pronounced when explicitly forcing tool use, where the accuracy drops significantly to 74.5%. These results imply that restricting agent behavior through either premature collapse or forced fixation could potentially compromise generalist capabilities.

In contrast, the inclusion of an entropy-regularization term yields the only approach that surpasses the base model, achieving 78.8% accuracy. We interpret this improvement as empirical evidence consistent with the hypothesis that diverse exploration during training may serve as a form of visual scaffolding. By encouraging the agent to broadly engage with various visual cues, the model appears to acquire more robust spatial understandings that possess better transferability to general depth tasks, rather than strictly overfitting to the tool-use patterns of the training dataset. Overall, these findings underscore the potential importance

of exploration diversity for enhancing both specific reasoning capabilities and general visual understanding.

### 4.3. Do Visual Agents Already Have Spatial Reasoning Ability?

**Partially Yes.** Tab. 3 suggests that tool-using visual agents already exhibit non-trivial spatial reasoning ability even without any explicit spatial training. Compared to the generalist baseline (Qwen2.5-VL 7B), both DeepEyes and Mini-o3 improve accuracy on 3DSRBench, which indicates that agentic tool use and visual chain-of-thought can recover fine-grained evidence through tool-mediated inspection that is sometimes missed from a single global view. At the same time, this advantage does not transfer uniformly. On CV-Bench-3D, DeepEyes underperforms the generalist baseline, and Mini-o3 shows a smaller margin, which highlights that tool use is not a guaranteed fix for spatial reasoning and that different spatial benchmarks can stress different failure modes.

A natural concern is that our gains could simply reflect starting from a strong spatial agent such as Mini-o3. The specialist reference SpatialReasoner provides a useful comparison here. SpatialReasoner reaches 60.3% on 3DSRBench by explicitly incorporating 3D coordinates into textual reasoning during its SFT-RFT pipeline. Our best model achieves 62.9% on 3DSRBench without using such explicit 3D representations or any dedicated spatial supervision in the initialization. This indicates that the main driver of improvement is not an unusually strong spatial pretraining recipe, but the training-time exploration dynamics studied in this work.

## 5. Conclusion

We investigated the role of visual tools in an "optional tool" regime, using 3D spatial reasoning as our main testbed. We identified a consistent *tool-use collapse* dynamic where tool use drops to near zero as accuracy improves, even under methods designed to mitigate length penalties. This collapse does not imply tools are unnecessary; banning them degrades performance, confirming that early interactions provide essential learning signals. However, forcing usage via rewards yields only marginal gains and homogenizes reasoning rollouts. Adaptive entropy regularization proves more effective, achieving the best performance while allowing tool use to fade naturally. Our quantitative analysis confirms that this approach prevents homogenized rollouts, sustaining broader exploration in both textual reasoning and visual tool invocation. Additional results on medical VQA suggest that this dynamic extends beyond 3D spatial reasoning and crop-based perception. These findings support a view of tools as *scaffolding*. Early tool access facilitates broad exploration of evidence during training, shaping robust representations that enable accurate inference even after explicit dependence fades.

## Acknowledgment

This research was supported by the Institute of Information & Communications Technology Planning & Evaluation (IITP) grant funded by the Korea government(MSIT) (No. RS-2019-II190079, Artificial Intelligence Graduate School Program(Korea University), 1%; RS-2025-02653113, High-Performance Research AI Computing Infrastructure Support at the 2 PFLOPS Scale, 1%; No. RS-2025-25439490, 70%), and the National Research Foundation of Korea(NRF) grant funded by the Korea government(MSIT)(No. RS-2025-02263628, 28%).

## Impact Statement

This work provides insights into the dynamics of tool use in visual agents, suggesting that diverse exploration during training may serve as effective scaffolding for robust representation learning. By reducing dependency on inference-time tools, our findings could contribute to the development of more efficient autonomous systems, though further research is warranted to ensure these internalized representations remain reliable across varied real-world contexts.

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

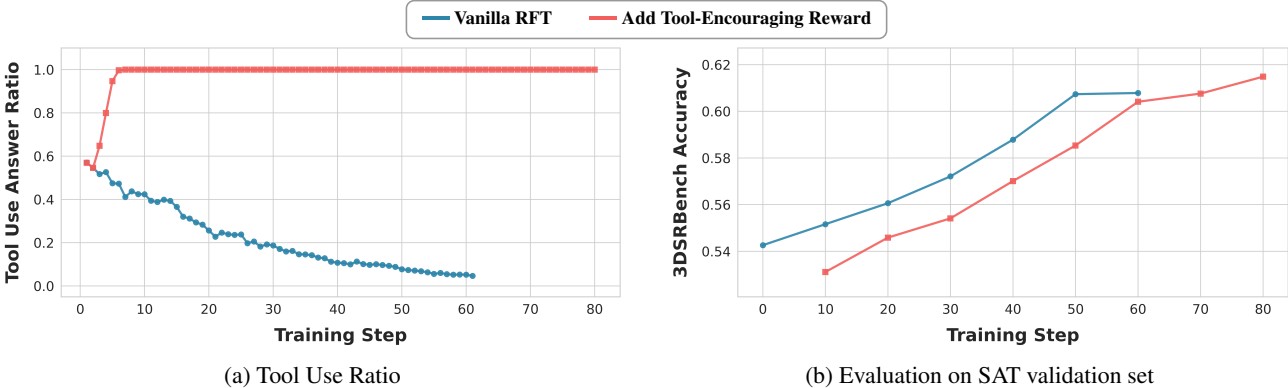

(a) Tool Use Ratio               (b) Evaluation on SAT validation set

*Figure A.* Analysis of Training Dynamics on SAT. (a) Tool Use Ratio over training defined as the fraction of rollouts that contain at least one grounding action. (b) Validation accuracy trends. Vanilla training (blue) and tool-encourage reward (red) exhibit divergent tool usage behaviors despite similar limited accuracy gains. This confirms that optimization pathologies persist regardless of dataset size.

## A. Related Works

### A.1. Visual Reasoning Agents via Visual Chain-of-Thought

Recent visual agents employ visual tools during chain-of-thought reasoning to enable fine-grained inspection. Most approaches follow a training pipeline of supervised fine-tuning followed by reinforcement learning with GRPO (Su et al., 2025; Wu et al., 2025; Zheng et al., 2026; Su et al., 2026; Lai et al., 2026; Zhu et al., 2025; Wang et al., 2025; Zhang et al., 2025a). These methods can be categorized by their tool repertoire. One category focuses exclusively on zoom-in and cropping operations. DeepEyes (Zheng et al., 2026), Mini-o3 (Lai et al., 2026), Active-o3 (Zhu et al., 2025), Chain-of-Focus (Zhang et al., 2025a), and PixelReasoner (Su et al., 2026) train models to adaptively invoke zoom or crop tools through bounding box coordinates, with Mini-o3 employing thought-action-observation loops spanning up to 32 turns. A second category incorporates diverse visual tool sets beyond zooming. OpenThinkIMG (Su et al., 2025) standardizes nine vision tools including object detection, segmentation, and OCR. Simple-o3 (Wang et al., 2025) demonstrates interleaved vision-language reasoning with cropping, zooming, and image reuse operations. VTool-R1 (Wu et al., 2025) generates multimodal chains of thought using Python-based visual editing tools, while ReFocus (Fu et al., 2025) employs Python code to draw boxes, highlight sections, and mask regions. Visual Sketchpad (Hu et al., 2024) enables test-time tool invocation through sketching and specialist vision models. While these methods achieve strong performance on visual reasoning benchmarks, the training dynamics of tool-use behavior remain underexplored. Our work specifically investigates how RL training shapes tool-use patterns in 3D spatial reasoning tasks where tool utility is ambiguous rather than mandatory, revealing that vanilla RL can drive tool usage toward collapse while exploration-based methods enable tools to function as scaffolding that supports robust tool-free inference.

### A.2. Analyzing Visual Chain-of-Thought Behavior

Beyond developing visual CoT methods, recent work has systematically investigated when and how explicit visual information within chain-of-thought reasoning benefits vision-language models. One line of work (Xu et al., 2025) demonstrates that while visual CoT with intermediate edited image patches consistently improves accuracy, it also increases sensitivity to input perturbations, with these intermediate visual reasoning components serving as the primary source of fragility. Another study (Du et al., 2025) investigates how different CoT designs affect generalizable visual reasoning ability, finding that visual CoT with intermediate image generation enables better generalization compared to text-only approaches after reinforcement learning. Our work complements these analyses by investigating the training dynamics of tool-mediated visual reasoning, revealing how exploration diversity determines whether explicit visual tools function as scaffolding or lead to usage collapse.

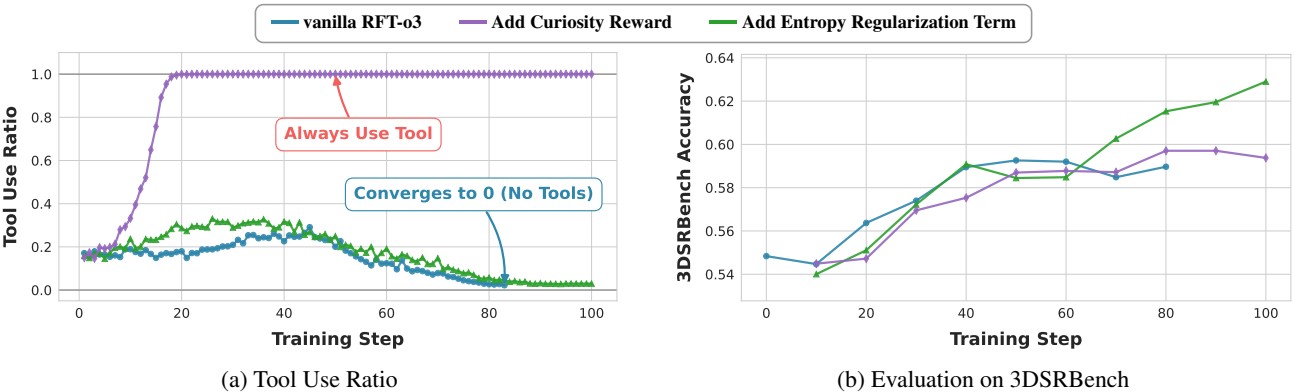

(a) Tool Use Ratio          (b) Evaluation on 3DSRBench

*Figure B.* **Analysis of Training Dynamics.** (a) Tool Use Ratio over training, defined as the fraction of rollouts that contain at least one `<grounding>` action. (b) Validation accuracy trends. Vanilla training (blue) and add curiosity reward (purple) exhibit divergent tool usage behaviors despite similar accuracy gains.

| Method | 3DSRBench Avg@8 |
|---|---|
| Vanilla RFT | 59.2 |
| Early-Only Entropy Reg | 62.5 |
| Full Entropy Reg | 62.9 |

*Table A.* Early-phase-only entropy regularization preserves most of the gains of full entropy regularization.

## B. Additional Experiments

### B.1. Training on SAT Dataset

We extend our analysis to the larger SAT dataset (Ray et al., 2025) to verify if the observed tool-use collapse is specific to the smaller 3DSRBench dataset. Figure A presents the training dynamics on SAT. The initial tool-use ratio starts significantly higher at approximately 40% compared to the main experiments. However, the overall trends remain consistent where Vanilla RFT exhibits tool-use collapse as the ratio steadily declines to near zero. Conversely, adding a tool-encouraging reward drives the agent to always use tools. This confirms that these optimization pathologies persist regardless of dataset size or the initial tool-use rate of the policy.

### B.2. Training on Add Curiosity Reward

We provide additional experimental results using the Curiosity Reward design (Su et al., 2026) discussed in the main text to verify if a more sophisticated reward design mitigates optimization pathology. Figure B illustrates the training dynamics comparing Vanilla RFT, Add Curiosity Reward, and Add Entropy Regularization Term. Consistent with the Tool Bonus results reported in the main text, the Curiosity Reward drives the agent to rapidly saturate tool usage. The tool-use ratio quickly escalates to near 100% within the first 20 training steps. While this approach effectively prevents tool-use collapse, it results in the opposite extreme of indiscriminately invoking tools for every query. Consequently, the validation accuracy shows only marginal improvements and plateaus at a lower level compared to the significant gains achieved by our adaptive entropy regularization method. This reinforces our conclusion that extrinsic reward engineering tends to force rigid tool-use behaviors rather than fostering the diverse exploration required for robust reasoning.

### B.3. Early-Phase Entropy Ablation

To more directly test the scaffolding interpretation, we apply entropy regularization only during the early phase of training. Specifically, we use the same entropy-regularized objective for the first 60 training steps and then disable the entropy term for the remainder of training.

As shown in Table A, applying entropy regularization only in the early phase already recovers most of the gain achieved by full entropy regularization. This supports our scaffolding interpretation: early exploratory pressure improves the learned policy, and the benefit largely persists even after the additional exploration pressure is removed.

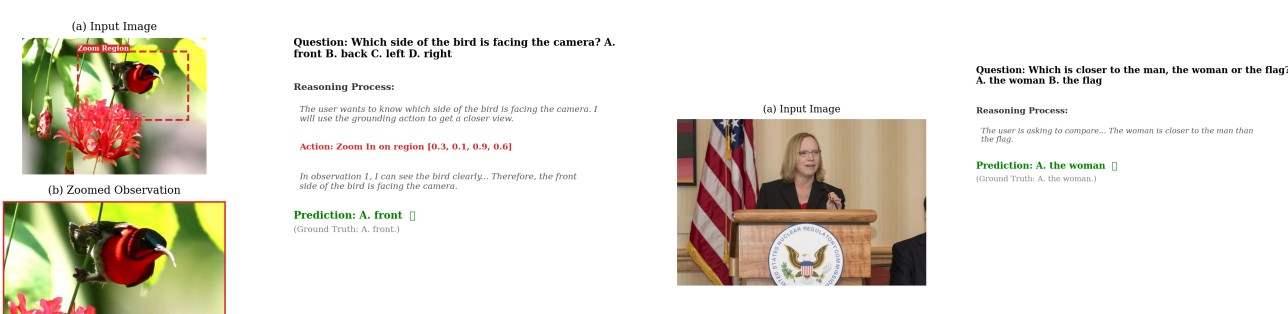

*(a)* Multi-Turn tool usage (Case with zoom-in)        *(b)* Direct Answer (Case without Grounding)

*Figure C.* Comparison of reasoning processes. (a) An example where the model performs a grounding action (zooming in) to identify the bird's orientation. (b) An example where the model answers directly based on the original image.

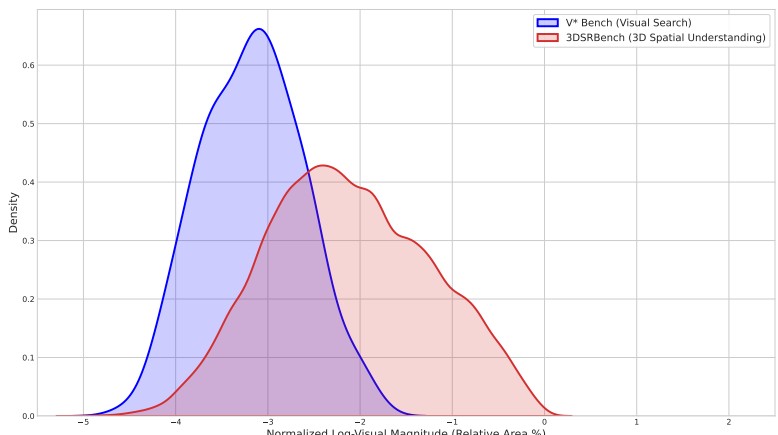

*Figure D.* Comparison of Target Object Scales. The figure plots the density distribution of normalized log-visual magnitude for target objects in V* (blue) (Wu & Xie, 2024) and 3DSRBench (red) (Ma et al., 2025a). While visual search tasks primarily focus on detecting extremely small objects, 3D spatial reasoning involves a broader range of object sizes. This indicates that the core challenge in 3D spatial reasoning lies not merely in detection but in understanding the relative spatial relationships among objects of various scales.

## C. Extra Visualization

### C.1. Tool-Use and No Tool Example

Figure C provides qualitative examples of model rollouts illustrating both multi-turn tool usage and direct tool-free inference.

### C.2. Benchmark Difference Between Visual Search and 3D Spatial Understanding Benchmarks

As illustrated in Figure D, visual search benchmarks are heavily skewed toward extremely small objects, implicitly making tool usage mandatory for detection. In contrast, 3D spatial understanding covers a wide spectrum of object scales, which necessitates the agent to learn a selective strategy to determine whether fine-grained inspection is required for relational reasoning or if the global view suffices.

### C.3. Token Length Evolution During Training

We analyze the average length of the `<think>` block to assess the depth of reasoning during training. The tool-encouraging reward leads to a progressive decline in token count which suggests the model shortcuts reasoning to greedily pursue tool usage. In contrast, adding the entropy regularization term induces a sustained increase in generation length which indicates that fostering exploration encourages the model to develop richer internal reasoning rather than converging to simplistic behaviors.

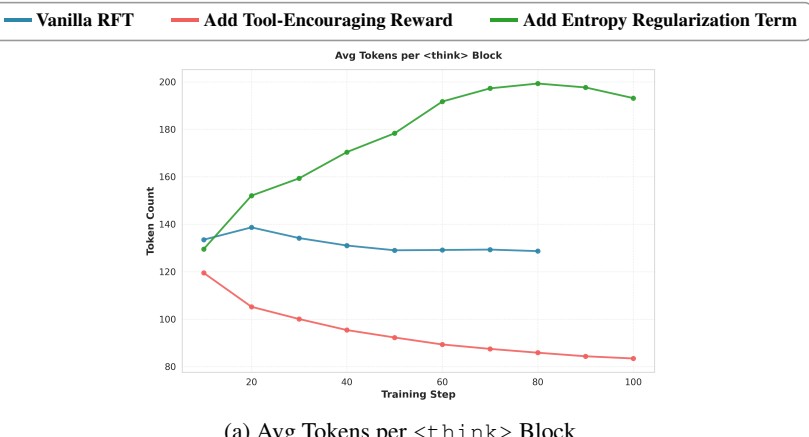

(a) Avg Tokens per `<think>` Block

*Figure E.* **Analysis of Training Dynamics.** (a) The average number of tokens within the `<think>` block across training steps. *Vanilla RFT* (blue) exhibits an initial increase followed by stabilization around 130 tokens. The *Tool-Encouraging Reward* (red) leads to a continuous decrease in the token count, suggesting shorter reasoning processes. In contrast, the *Entropy Regularization Term* (green) encourages exploration, resulting in a sustained increase in the length of the `<think>` block.

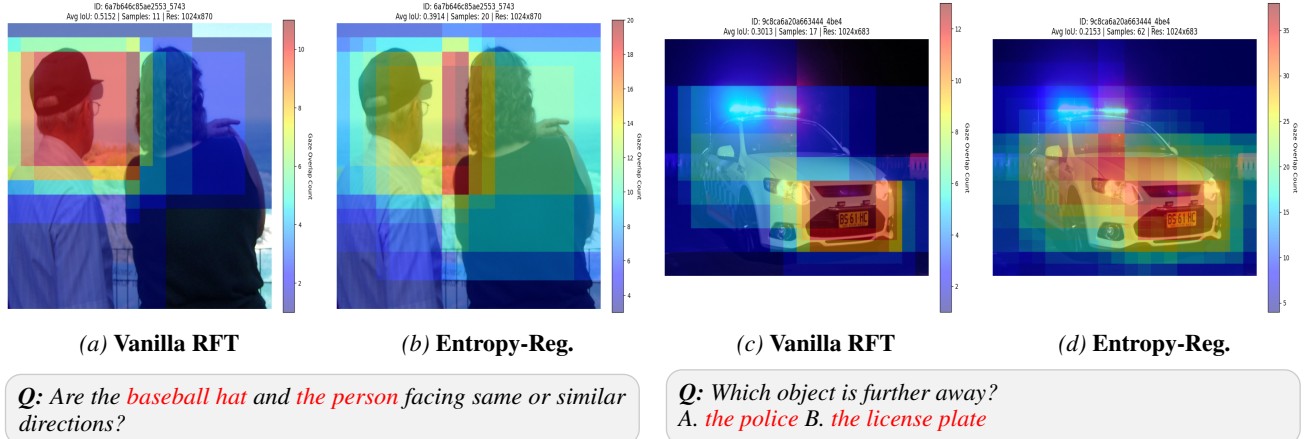

*(a)* **Vanilla RFT**      *(b)* **Entropy-Reg.**      *(c)* **Vanilla RFT**      *(d)* **Entropy-Reg.**

*Q:* Are the *baseline hat* and *the person* facing same or similar directions?

*Q:* Which object is further away?
A. *the police* B. *the license plate*

*Figure F.* **Additional Visual Exploration During Training Examples.** Comparison of crop distributions. The baseline models (a, c) focus narrowly, while entropy-regularized models (b, d) explore contextually relevant regions.

## C.4. Additional Visual Exploration Visualization

Figure F presents additional qualitative comparisons of crop heatmaps between the vanilla RFT baseline and the entropy-regularized model during early training. These examples further demonstrate that the entropy-regularized policy maintains broader visual exploration, effectively attending to contextually relevant peripheral regions, whereas the baseline tends to collapse prematurely into a narrow focus on the primary subject.

## D. VLM-as-Judge Reliability

As described in Section 2.2, we evaluate free-form model outputs on 3DSRBench using a fixed VLM-as-judge (Qwen2.5-VL-7B-Instruct), which maps each response to one of the multiple-choice options before comparing it with the ground-truth label. To validate the reliability of this automatic evaluation protocol, we additionally conduct a human verification study on 384 samples, with 128 samples independently annotated for each condition: Vanilla RFT, Tool-Encouraged RFT, and Entropy Regularization.

Table B shows that the overall mismatch rate is 4.4%, indicating that the automatic judge is largely consistent with human verification. Moreover, because 3DSRBench is a multiple-choice benchmark, the role of the judge is limited to answer-option extraction rather than open-ended semantic evaluation, which reduces the opportunity for systematic bias. These results support the reliability of the VLM-as-judge protocol used throughout the paper.

| Condition | Samples Judged | Judge-Human Mismatch |
|---|---|---|
| Vanilla RFT | 128 | 2.3% |
| Tool-Encouraged RFT | 128 | 6.25% |
| Entropy Reg. | 128 | 5.47% |
| Overall | 384 | 4.4% |

*Table B.* Agreement between the automatic VLM-as-judge and human verification.

