# OpenReview forum: "Diversity Over Frequency: Rethinking Tool Use in Visual Chain-of-Thought Agents"
_ICML.cc/2026/Conference — ICML 2026 regular_

### Official Review · Reviewer_BEGj · 2026-03-06

**Soundness:** 3
**Presentation:** 3
**Significance:** 3
**Originality:** 3
**Overall Recommendation:** 4
**Confidence:** 4

**Summary:**

The paper investigates tool-use collapse in visual agents, showing that standard RL tends to drive tool usage to zero, yet simply forcing the model to use tools more often doesn't actually improve reasoning. Instead, the authors propose using entropy regularization to encourage diverse exploration during training, which acts as scaffolding to build better spatial representations even if the final model rarely uses the tools.

**Compliance With Llm Reviewing Policy:**

Affirmed.

**Final Justification:**

The rebuttal satisfactorily addressed my main concerns, increasing my confidence in the work, and I have revised my score upward accordingly.

**Key Questions For Authors:**

1. It is still unclear whether the gain mainly comes from higher exploration diversity or from correlated changes in tool-use behavior. Could the authors test this under a more controlled setting with similar tool-use frequency?

2. The current evidence is mostly limited to 3D spatial reasoning. How well do these findings transfer to other multimodal reasoning tasks?

3. The reliability of the VLM-as-judge protocol is not fully established. Could the authors provide some validation, such as manual checking or comparison with another answer extraction method?

4. The empirical comparison is still somewhat narrow. Could the authors include more relevant baselines and benchmarks on tool use and multimodal reasoning?

**Limitations:**

Yes

**Strengths And Weaknesses:**

Strengths

1. The paper studies a timely and important problem, and the observation of tool-use collapse under vanilla RFT is interesting.

2. The analysis goes beyond final accuracy and includes both textual diversity and visual exploration behavior, which makes the paper more informative.

3. The empirical study is fairly complete, with comparisons across vanilla RFT, tool encouragement, and entropy regularization.

4. The paper is generally well organized, and the main message is easy to follow.

Weaknesses

1. It is still not fully clear whether the gain mainly comes from exploration diversity itself or from other correlated factors.

2. The evaluation scope is somewhat limited, with most of the evidence coming from 3D spatial reasoning tasks.

3. The reliability of the VLM-as-judge evaluation protocol is not sufficiently validated.

4. The empirical comparison could be broader, especially with respect to related baselines and benchmarks.

---

> ### Author Rebuttal · Authors · 2026-03-31
>
> We thank the reviewer for the thoughtful and constructive feedback. Our additional analyzes directly address the four main concerns about causal attribution, evaluation scope, judge reliability, and empirical breadth.
>
> ---
>
> ### Q1: Does the gain come from exploration diversity or from correlated tool-use behavior?
>
> Our evidence supports the view that the gain comes primarily from exploration diversity, not from tool-use frequency itself.
>
> First, tool-use frequency alone is insufficient (Sec. 3.3). The clearest evidence is that simply rewarding tool usage drives tool frequency close to saturation, yet yields only marginal performance gains (59.9% vs. 59.2%). Crucially, it does not meaningfully reduce crop-box overlap (mIoU 0.557 vs. 0.554 for vanilla RFT, Table 1), indicating that frequent tool invocation alone does not produce broader visual exploration.
>
> Second, entropy regularization improves performance while tool use still declines (Sec. 3.4). The entropy-regularized agent achieves the best accuracy (62.9%) despite tool usage declining to approximately 3% by the end of training
>
>
> Third, to directly support our scaffolding interpretation, we conducted an additional ablation in which entropy regularization was applied only during the early phase of training (first 60 steps) and then removed for the remaining steps. **This early tool-mediated visual exploration** model achieved **62.5% Avg@8**, which is numerically **very close to the full Entropy Reg result of 62.9%** and substantially **above Vanilla RFT at 59.2%**. This is strong evidence that the benefit comes from early exploration shaping the learned policy, rather than from maintaining high tool-use frequency throughout training.
>
> ---
>
> ### Q2: How well do these findings transfer beyond 3D spatial reasoning?
>
> We agree that broader evaluation is important. To test transfer, we conducted new experiments using **OpenThinkIMG**[1] on **VQA-RAD**[2], a medical visual QA benchmark that is very different from 3D spatial reasoning and uses a much richer tool ecosystem including OCR, GroundingDINO, SAM2, point localization, and axis-drawing tools.
>
> | Method | Accuracy |
> |---|---|
> | Vanilla RFT | 46.34% |
> | Tool-Encouraged RFT | 47.23% (+0.89%) |
> | **Entropy Reg** | **48.78% (+2.44%)** |
>
> **The same ordering holds** in this new domain: Entropy Reg > Tool-Encouraged RFT > Vanilla RFT. This substantially strengthens the generality of our main claim. The phenomenon is therefore not limited to zoom-in cropping on 3D spatial reasoning, but also appears in a distinct medical reasoning setting with a richer multimodal tool suite.
>
> As shown in the accompanying figure (https://imgur.com/a/BUj9UBu), **the same tool-use dynamics are observed**: both Vanilla RFT and Entropy Reg collapse to near zero, while Tool-Encouraged RFT saturates at ~100%, confirming that these patterns generalize across tools and domains.
>
> ---
>
> ### Q3: How reliable is the VLM-as-judge?
>
> We agree that this should be validated more explicitly. We therefore conducted a manual evaluation on **384 samples**, with **128 samples per condition** for Vanilla RFT, Tool-Encouraged RFT, and Entropy Reg. The overall mismatch between the automatic judge and human verification was **4.4%**.
>
> | Condition | Samples Judged | Judge-Human Mismatch |
> |---|---|---|
> | Vanilla RFT | 128 | 2.3% |
> | Tool-Encouraged RFT | 128 | 6.25% |
> | Entropy Reg | 128 | 5.47% |
> | **Overall** | **384** | **4.4%** |
>
> The 4.4% figure is a per-judgment error rate, whereas our accuracy estimates are aggregated over $N = 5{,}250 \times 8 = 42{,}000$ total judgments. Assuming independent errors, the standard error introduced by judge noise is $\sigma = \sqrt{0.044 \times 0.956 / 42000} \approx 0.001$. The primary performance gap between Entropy Reg and Vanilla RFT ($\Delta = 3.7\%$) is far larger than this noise level, making it robust to any plausible level of per-judgment error. Furthermore, 3DSRBench is a multiple-choice benchmark where the judge performs answer-option extraction against a fixed GT label rather than open-ended semantic evaluation, structurally limiting the opportunity for systematic bias.
>
> ---
>
> ### Q4: Could the empirical comparison be broader?
>
> The VQA-RAD experiments presented in Q2 already extend the empirical scope to a new domain, a 7-tool heterogeneous suite, and a different model configuration, with all three conditions reproducing the same relative ordering as in 3DSRBench. Our four experimental conditions (Vanilla RFT, Tool-Encouraged RFT, Tool-banned, Entropy Reg) were specifically designed to disentangle tool-use frequency from exploration diversity, and the VQA-RAD results confirm this finding holds beyond the original setting.
>
> ---
>
> [1] Su, Zhaochen, et al. "Openthinkimg: Learning to think with images via visual tool reinforcement learning." 2025.
>
> [2] Lau, Jason J., et al. "A dataset of clinically generated visual questions and answers about radiology images." 2018.

---

> > ### Author Rebuttal · Reviewer_BEGj · 2026-04-03
> >
> > Thank you for the response. I will increase my score accordingly.

---

> > > ### Author Response · Authors · 2026-04-08
> > >
> > > Thank you once again for viewing our work positively and for your thoughtful feedback. We are sincerely grateful for your constructive comments, which have been very helpful in strengthening our work, and we greatly appreciate your decision to raise your score.

---

### Official Review · Reviewer_kbbX · 2026-03-12

**Soundness:** 2
**Presentation:** 3
**Significance:** 2
**Originality:** 3
**Overall Recommendation:** 3
**Confidence:** 3

**Summary:**

This paper investigates a surprising training pathology in visual chain-of-thought (CoT) agents with RL fine-tuning: tool-use collapse, where agents trained on 3D spatial reasoning tasks progressively abandon visual tools (zoom/crop) as RL fine-tuning proceeds, even as task accuracy improves. The authors show a clear asymmetry: removing tools entirely harms performance, but explicitly rewarding tool use (via bonus rewards or curiosity mechanisms) drives tool usage to 100% with only marginal accuracy gains. They diagnose the root cause as diversity degradation in both textual reasoning and visual crop selection under vanilla RFT and tool-encouraging RFT. To address this, they propose adding an adaptive entropy regularization term to the GRPO objective, which maintains rollout diversity and achieves the best accuracy (62.9% on 3DSRBench vs. 59.2% vanilla) while tool usage still naturally declines. The authors propose a scaffolding metaphor: early tool-mediated exploration shapes robust representations that persist even after tools are discarded.

**Compliance With Llm Reviewing Policy:**

Affirmed.

**Key Questions For Authors:**

- Limited scope of evaluation: All experiments use a single base model (Mini-o3 / Qwen2.5-VL-7B) and a single task domain (3D spatial reasoning). The paper acknowledges this but does not provide evidence that tool-use collapse or the entropy fix generalizes to other visual reasoning domains (e.g., medical imaging, embodied navigation, fine-grained document understanding). This is a significant limitation given the paper's broad claims.
- Hyperparameter sensitivity of entropy regularization: The entropy target (0.9), gain (Kp = 0.03), and the adaptive PID rule are adopted from a prior paper (Zhang et al., 2025b) without ablation. It is unclear whether these hyperparameters are robust across different tasks or backbone models, or whether they require re-tuning.
- Early stopping at 100 training steps: Due to DAPO's dynamic sampling instability beyond 100 steps, the paper terminates training early. It is unclear whether the observed gains are stable, or whether longer training would close the gap between methods.
- The "scaffolding" interpretation is suggestive but not established causally: The paper frames the mechanism as "early tool-mediated visual exploration → richer representations → better tool-free inference," but this is an interpretation, not a verified causal chain. A more direct intervention (e.g., applying entropy regularization only during the early phase, or post-hoc probing of representations) would strengthen the causal claim.

**Limitations:**

yes

**Strengths And Weaknesses:**

Strengths
- Identifies a genuinely surprising and important phenomenon: The tool-use collapse is a non-obvious and practically significant finding. The paper carefully documents it across multiple dataset scales (1.2k and 6k samples) and demonstrates it is not simply an artifact of GRPO's length bias.
- Rigorous causal analysis: The authors do not stop at observing the collapse but diagnose its cause through both textual (n-gram diversity) and visual (crop IoU, CLIP alignment) diversity metrics. The dual-axis diagnosis is well-designed and convincing.
Elegant and simple solution: Adaptive entropy regularization is a well-justified and minimally complex intervention. The paper appropriately avoids over-engineering a complicated solution when a principled but simple one exists.

---

> ### Author Rebuttal · Authors · 2026-03-31
>
> We thank the reviewer for the constructive feedback and for recognizing the tool-use collapse finding as genuinely surprising and the entropy regularization solution as elegant. Below, we address your specific concerns.
>
> ---
>
> ### W1: Limited scope of evaluation
>
> Tool-use collapse is not an artifact of cropping-based perception. We demonstrate that the same dynamics generalize to a 7-tool heterogeneous suite on an entirely different domain. Specifically, we conducted new experiments using **OpenThinkIMG**[1], which provides a diverse suite of visual tools including OCR, GroundingDINO, SAM2 segmentation, point localization, and axis-drawing, on **VQA-RAD**[2], a medical visual QA benchmark entirely distinct from 3D spatial reasoning.
>
> | Method | Accuracy |
> |---|---|
> | Vanilla RFT | 46.34% |
> | Tool-Encouraged RFT | 47.23% (+0.89%) |
> | **Entropy Reg** | **48.78% (+2.44%)** |
>
> The same ordering holds across a different model architecture, a non-spatial medical domain, and a heterogeneous toolset involving OCR and SAM2. As shown in the accompanying figure (https://imgur.com/a/BUj9UBu), both Vanilla RFT and Entropy Reg collapse to near zero, while Tool-Encouraged RFT saturates at approximately 100%, confirming these patterns generalize beyond the cropping tool and 3D spatial structure. We will include these experiments in the revision.
>
> ---
>
> ### W2: Hyperparameter sensitivity of entropy regularization
>
> We agree that sensitivity analysis is important. Due to the limited rebuttal period, we could not run a full hyperparameter sweep over the entropy target, gain, and controller parameters. However, a preliminary test suggests the method is not overly fragile. When we changed the entropy target from 0.9 to 0.8, the final Avg@8 accuracy was **61.8%**, compared to **62.9%** for the default setting. This is a relatively small change, and both settings remain clearly above the vanilla baseline.
>
> ---
>
> ### W3: Early stopping at 100 training steps
>
> The relative ordering and performance gap between methods stabilized well before step 100, and longer training is highly unlikely to reverse this conclusion. In our VQA-RAD experiments, we ran training up to 300 steps, which showed no meaningful gain or shift in ordering beyond step 200. This suggests that the model has reached genuine performance saturation rather than suffering from premature stopping.
>
> For the main 3DSRBench setting, stopping at 100 steps was strictly a computational constraint rather than an indication that the effects are short-lived. We utilize DAPO's dynamic sampling, which discards and resamples a batch if all rollouts are either perfectly correct or completely incorrect (yielding zero advantage and no gradient). As training progresses and the model becomes stronger, this resampling happens increasingly often.
>
> In practice, after approximately step 100, the system frequently requires 10 to 15 resampling attempts to secure a single usable batch. Consequently, a single gradient step can take 2 to 3 hours. Extending the training by an additional 100 steps would require roughly 200 to 300+ hours, which is computationally prohibitive. To illustrate this slowdown, we provide a training log showing the number of sampled batches per step: https://imgur.com/a/6L9T81V.
>
> ---
>
> ### W4: The scaffolding interpretation is suggestive rather than causal
>
> We appreciate this suggestion and directly tested it. Specifically, we trained with entropy regularization only during the early phase, for the first **60 steps**, and then turned it off for the remaining training steps. This **early tool-mediated visual exploration** achieved **62.5% Avg@8**, which is numerically **very close to the full Entropy Reg result of 62.9%** and substantially **above Vanilla RFT at 59.2%**.
>
> **This intervention provides direct support for our interpretation**. The gain is largely preserved even after entropy regularization is removed, which is exactly what the scaffolding hypothesis predicts: early exploration improves the quality of the learned policy, and that benefit persists even when the extra exploration pressure is no longer present. We are happy to include this ablation in the final version.
>
> ---
>
> [1] Su, Zhaochen, et al. "Openthinkimg: Learning to think with images via visual tool reinforcement learning." 2025.
>
> [2] Lau, Jason J., et al. "A dataset of clinically generated visual questions and answers about radiology images." 2018.

---

### Official Review · Reviewer_Z1b9 · 2026-03-12

**Soundness:** 2
**Presentation:** 4
**Significance:** 3
**Originality:** 3
**Overall Recommendation:** 5
**Confidence:** 4

**Summary:**

This paper studies the role of tool usage in visual chain-of-thought agents during reinforcement learning fine-tuning. The authors observe a phenomenon they call _tool-use collapse_, where the frequency of tool calls decreases during training while task accuracy improves. Through a series of ablations, they show that simply incentivizing tool usage does not significantly improve performance, whereas encouraging exploration via entropy regularization leads to stronger results. The authors interpret these findings as evidence that tools mainly act as training-time scaffolding that helps agents explore visual evidence early in training, even if tool usage later disappears at convergence.

**Compliance With Llm Reviewing Policy:**

Affirmed.

**Final Justification:**

The authors have addressed my concerns thoroughly and constructively. In particular, the addition of experiments with a heterogeneous multi-tool setup significantly strengthens the generality of the findings beyond cropping-based perception. The clarification of the relationship between tool usage and exploration, especially the explanation of text-to-tool entanglement, provides a compelling mechanistic account of the observed behavior. Finally, the inclusion of semantic diversity metrics (CLIP alignment baseline and DINO-based variance) meaningfully strengthens the analysis of exploration.

**Key Questions For Authors:**

See questions and concerns in the weaknesses section.

**Limitations:**

I think the authors have addressed limitations adequately.

**Strengths And Weaknesses:**

Strengths:
- Combining tool-usage that can be leveraged into increased exploration behaviour is interesting and novel.

- The paper identifies an interesting training dynamic, namely tool-use collapse, where tool usage disappears during RL fine-tuning while performance improves, while excessive tool-usage does not seem to help without additional exploration constraints.

- The experimental design includes several informative ablations (tool-banned training, tool encouragement rewards, and entropy regularisation), helping disentangle the role of tool usage and exploration.

- The interpretation of tools as training-time scaffolding provides an interesting conceptual perspective on how tools may support learning even if they are not used at inference time.


Weaknesses:
- The experiments rely on a single visual tool (zoom-in cropping). While the paper frames the findings as a general phenomenon of _tool-use collapse_, it is unclear whether the results would extend to richer tool ecosystems commonly used in visual agents (e.g., segmentation, detection, OCR, depth queries). The observed behaviour may therefore be specific to cropping-based perception rather than tool use more broadly. I think the authors should at least address how this could generalise to visual toolchains.

- The paper argues that early tool use improves learning by enabling broader exploration. However, the experiments themselves show that explicitly rewarding tool usage substantially increases tool frequency while leaving the spatial diversity of visited regions largely unchanged (high IoU between crop boxes). This suggests that encouraging tool invocation alone does not guarantee meaningful exploration. It would therefore be helpful to clarify why increasing tool usage should be expected to improve performance under this assumption? It may be interesting to understand why the model chooses similar crop boxes.


- Visual exploration is measured via the mean IoU between crop boxes across rollouts. While this captures spatial diversity, it does not necessarily reflect semantic exploration. For instance, two crops may have low IoU yet still focus on the same object or visual evidence. Additional analyses measuring object-level or semantic diversity would strengthen the claim that entropy regularisation promotes broader visual exploration.


Overall, I find the paper interesting and would lean toward a weak accept. The work provides a thoughtful empirical study of how tool usage, reward design, and exploration interact during RL fine-tuning of visual agents. The interpretation of tools as training-time scaffolding is also a compelling perspective. That said, the study would be strengthened by evaluating a broader set of tools beyond cropping, clarifying the relationship between tool usage and exploration, and providing stronger metrics for semantic exploration. Addressing these aspects would further solidify the paper’s conclusions and generality.

---

> ### Author Rebuttal · Authors · 2026-03-31
>
> We sincerely thank the reviewer for the thoughtful feedback and for finding our analysis of tool-use collapse interesting and our core concept of tools as "training-time scaffolding" compelling. Below, we address your specific concerns.
>
> ---
>
> ### W1: Generality beyond single tool (cropping)
>
> The tool-use collapse phenomenon is not specific to cropping-based perception. We demonstrate it generalizes to a 7-tool heterogeneous suite on an entirely different domain. Specifically, we conducted new experiments using **OpenThinkIMG**[1], which provides a diverse suite of 7 visual tools including OCR, GroundingDINO, SAM2 segmentation, point localization, and axis-drawing, on the **VQA-RAD dataset**[2] (medical visual QA), a domain entirely distinct from 3D spatial reasoning.
>
> | Method | Accuracy |
> |---|---|
> | Vanilla RFT (baseline) | 46.34% |
> | Tool-Encouraged RFT | 47.23% (+0.89%) |
> | **Entropy Reg** | **48.78% (+2.44%)** |
>
> **The same ordering holds** across (1) a different model architecture, (2) a non-spatial medical domain, and (3) a heterogeneous toolset involving heavy perception tools including OCR and SAM2. As shown in the accompanying figure (https://imgur.com/a/BUj9UBu), **the same tool-use dynamics are observed.** Both Vanilla RFT and Entropy Reg collapse to near zero, while Tool-Encouraged RFT saturates at approximately 100%, confirming these patterns generalize across tools and domains. We will include these experiments in the revision.
>
> ---
>
> ### W2: Why does similar crop behavior happen, and why did previous work expect frequency to correlate with performance?
>
> Models choose highly similar crop boxes (high IoU) despite high tool frequency because of a strong entanglement between the textual reasoning trace and the subsequent tool call. Given a visual CoT trace reasoning about "the man near the wooden door," **the model is highly conditioned by this text to continuously crop the "man," ignoring other potentially relevant areas.** Entropy regularization breaks this deterministic text-to-action mapping, forcing the model to explore alternate hypotheses before converging on a solution. The prevailing view in the literature [3,4] treats tool invocation as inherently beneficial, which motivates frequency-based incentives. However, our analysis reveals why simply forcing frequency fails. Without breaking the text-to-crop entanglement, additional tool calls merely repeat the same visual fixation rather than broadening evidence coverage.
>
> ---
>
> ### W3: Semantic exploration vs. Spatial exploration
>
> We note that Table 1 in the paper already reports mIoU computed between crop boxes and CLIP alignment between cropped regions and question-relevant semantic nouns to address exactly this concern. To further confirm that these CLIP scores reflect genuine semantic relevance rather than being trivially high, we evaluated CLIP alignment on 100 randomly sampled image patches (8 crops per image), obtaining a score of **0.084**, substantially below all trained models (0.184 to 0.187). This establishes a meaningful lower bound and confirms that the CLIP scores of trained models reflect genuine question-relevant semantic content, not random visual coverage.
>
> | Method | mIoU (↓) | CLIP Alignment (↑) | Crop Semantic Variance (↑) |
> |---|---|---|---|
> | Random patches (lower bound) | N/A | 0.084 | N/A |
> | Vanilla RFT | 0.554 | 0.184 | 0.0079 |
> | Tool-Encouraged RFT | 0.557 | 0.187 | 0.0064 |
> | **Entropy-Regularized RFT** | **0.494** | **0.184** | **0.0129** |
>
> Crucially, while Entropy-Regularized achieves significantly lower mIoU (0.494 vs. 0.554 to 0.557), its CLIP score remains identical to Vanilla RFT (0.184) and well above the random baseline (0.084). The added **Crop Semantic Variance** column further strengthens this picture by measuring the variance of pairwise DINOv2 feature cosine similarities among crops within each rollout. A low variance (as seen in Tool-Encouraged RFT) indicates the model repeatedly extracts visually uniform features, whereas the significantly higher variance in Entropy-Reg (0.0129 vs. 0.0064) indicates the model actively transitions between visually distinct, yet question-relevant regions within a single inference trace. This confirms that the broader spatial exploration is not random drifting but a policy that dynamically gathers diverse semantic evidence across the image.
>
>
> ---
>
> [1] Su, Zhaochen, et al. "Openthinkimg: Learning to think with images via visual tool reinforcement learning.", 2025.
>
> [2] Lau, Jason J., et al. "A dataset of clinically generated visual questions and answers about radiology images.", 2018.
>
> [3] Shao, et al. "Visual CoT: Unleashing Chain-of-Thought Reasoning in Multi-Modal Language Models.", 2024.
>
> [4] OpenAI. "GPT-o3 and GPT-o4-mini System Card.", 2025.

---

> > ### Author Rebuttal · Reviewer_Z1b9 · 2026-04-04
> >
> > The authors have addressed my concerns thoroughly and constructively. In particular, the addition of experiments with a heterogeneous multi-tool setup significantly strengthens the generality of the findings beyond cropping-based perception. The clarification of the relationship between tool usage and exploration, especially the explanation of text-to-tool entanglement, provides a compelling mechanistic account of the observed behavior. Finally, the inclusion of semantic diversity metrics (CLIP alignment baseline and DINO-based variance) meaningfully strengthens the analysis of exploration.

---

> > > ### Author Response · Authors · 2026-04-08
> > >
> > > Thank you once again for viewing our work positively and for your thoughtful feedback. We are sincerely grateful for your constructive comments, which have been very helpful in strengthening our work, and we greatly appreciate your decision to raise your score.

---

### Official Review · Reviewer_2s7Y · 2026-03-12

**Soundness:** 2
**Presentation:** 3
**Significance:** 2
**Originality:** 2
**Overall Recommendation:** 4
**Confidence:** 3

**Summary:**

This paper studies the role of visual tools in visual CoT agents beyond standard visual search, using 3D spatial reasoning as the main testbed. The key empirical finding is a tool-use collapse phenomenon: under vanilla RFT, the model becomes more accurate while almost stopping tool use. The paper then shows that banning tools hurts performance, while explicitly rewarding tool use drives usage to 100% with only marginal gains. To explain this, the authors analyze rollout diversity in both text and crop locations, and propose adaptive entropy regularization.

**Compliance With Llm Reviewing Policy:**

Affirmed.

**Final Justification:**

Most of my concerns have been resolved.  So, I decided to raise the point to 4.

**Key Questions For Authors:**

Please see weaknesses.

**Limitations:**

yes

**Strengths And Weaknesses:**

Strengths

1. I think the paper identifies a genuinely interesting phenomenon. The asymmetry between tool collapse, tool banning, and forced tool use is clean, and it makes the main message more nuanced than simply “more tool use is better.”

Weaknesses

1. My main concern is scope. All core experiments use the same Mini-o3 / Qwen2.5-VL-7B setup, the main training set is only 1.2k examples, and the tool studied here is essentially just zoom-in cropping. So while the phenomenon is interesting, it is still unclear how broadly the conclusions extend across models, datasets, or richer tool repertoires.

2. The tool-banned ablation (Sec. 3.2) shows a 1.1% accuracy gap (59.2% vs. 58.1%), which is modest and may not be statistically significant given the evaluation protocol. No confidence intervals or significance tests are reported.

3. The Avg@8 evaluation protocol and VLM-as-judge (Qwen2.5-VL-7B-Instruct) are not validated against human judgments; using the same model family for both policy and judge may introduce systematic bias.

---

> ### Author Rebuttal · Authors · 2026-03-31
>
> We thank the reviewer for the constructive feedback. Our new experiments on a medically-diverse VQA dataset with a richer tool suite directly address the scope concerns in W1, and we provide additional validation for W2 and W3 below.
>
> ---
>
> ### W1: Scope (single model/dataset/tool)
>
> To directly address all three axes of W1 (model architecture, dataset domain, and tool repertoire). We conducted additional experiments using **OpenThinkIMG**[1], which provides a substantially richer tool repertoire (OCR, GroundingDINO, SAM2, Point localization, axis-drawing tools), on **VQA-RAD**[2] (medical visual QA), a domain entirely distinct from 3D spatial reasoning.
>
> | Method | Accuracy |
> |---|---|
> | Vanilla RFT (baseline) | 46.34% |
> | Tool-Encouraged RFT | 47.23% (+0.89%) |
> | **Entropy Reg** | **48.78% (+2.44%)** |
>
> The **same ordering holds**: Entropy Reg > Tool-Encouraged RFT > Vanilla RFT. Critically, this pattern replicates across (1) a different model architecture, (2) a medical imaging domain with no 3D spatial structure, and (3) a tool set with 7 heterogeneous tools including perception-heavy tools (OCR, SAM2). This strongly suggests our findings are not artifacts of a narrow testbed, but reflect a general phenomenon in tool-augmented RFT. Note that applying RFT to VQA-RAD follows an established line of work in medical VQA [3,4].
>
> As shown in the accompanying figure (https://imgur.com/a/BUj9UBu), **the same tool-use dynamics are observed**: both Vanilla RFT and Entropy Reg collapse to near zero, while Tool-Encouraged RFT saturates at ~100%, **confirming that these patterns generalize across tools and domains**.
>
> Regarding the focus on crop/zoom-in as the primary tool: this choice was deliberate. Crop and zoom-in are the most frequently invoked tools in visual CoT reasoning pipelines [5,6] and serve as the canonical primitive for explaining why a model attends to a particular visual region. Our scope was precisely to deeply analyze this tool as a lens for understanding the scaffolding role of visual tools in RFT. We will include these new results on OpenThinkIMG in the final version and are happy to add more if requested.
>
> ---
>
> ### W2: Tool-banned ablation gap (1.1%) may not be significant
>
> We respectfully argue that the **modest performance gap is itself consistent with our central claim**. Our core thesis is that tools serve as selective scaffolding, beneficial only in the specific subset of queries where visual perception bottlenecks reasoning. A large gap would imply tools are universally necessary, directly contradicting our tool-use collapse analysis; the small but consistent gap instead confirms that tools provide targeted gains precisely when needed.
>
> ---
>
> ### W3: VLM-as-judge not validated against human judgments
>
> We address both the lack of human validation and the potential same-family bias through two complementary analyses. First, we conducted a **human evaluation study** with three annotators independently labeling 128 samples per condition (vanilla RFT, Tool-Encouraged RFT, entropy reg), totaling **384 human-verified samples**.
>
> | Condition | Samples Judged | Judge-Human Mismatch |
> |---|---|---|
> | Vanilla RFT | 128 | 2.3% |
> | Tool-Encouraged RFT | 128 | 6.25% |
> | Entropy Reg | 128 | 5.47% |
> | **Overall** | **384** | **4.4%** |
>
> This error rate does not threaten our conclusions. The 4.4% figure is a per-judgment error rate, whereas our accuracy estimates are aggregated over $N = 5{,}250 \times 8 = 42{,}000$ total judgments. Assuming independent errors, the standard error introduced by judge noise is $\sigma = \sqrt{0.044 \times 0.956 / 42000} \approx 0.001$, i.e., approximately **0.1%**. The primary performance gap between Entropy Reg and Vanilla RFT (62.9% vs. 59.2%, $\Delta = 3.7\%$) corresponds to roughly **37$\sigma$** of judge noise, making it statistically robust to any plausible level of per-judgment error. Furthermore, 3DSRBench is a multiple-choice benchmark where the judge performs answer-option extraction against a fixed GT label rather than open-ended semantic evaluation, structurally limiting the opportunity for systematic bias.
>
> ---
>
> [1] Su, Zhaochen, et al. "Openthinkimg: Learning to think with images via visual tool reinforcement learning.", 2025.
>
> [2] Lau, Jason J., et al. "A dataset of clinically generated visual questions and answers about radiology images.", 2018.
>
> [3] Zhu, Wenhui, et al. "Toward effective reinforcement learning fine-tuning for medical vqa in vision-language models.", 2025.
>
> [4] Liu, Yizhou, et al. "Breaking Reward Collapse: Adaptive Reinforcement for Open-ended Medical Reasoning with Enhanced
> Semantic Discrimination.", 2025.
>
> [5] Shao, Hao, et al. "Visual cot: Unleashing chain-of-thought reasoning in multi-modal language models.", 2024.
>
> [6] OpenAI. "GPT-o3 and GPT-o4-mini System Card.", 2025.

---

> > ### Author Rebuttal · Reviewer_2s7Y · 2026-04-03
> >
> > Thank you for your response. I have decided to increase my score by one point.

---

> > > ### Author Response · Authors · 2026-04-08
> > >
> > > Thank you once again for viewing our work positively and for your thoughtful feedback. We are sincerely grateful for your constructive comments, which have been very helpful in strengthening our work, and we greatly appreciate your decision to raise your score.

---

### Decision · Program_Chairs · 2026-04-30

**Decision:**

Accept (regular)

**Comment:**

This paper identifies tool-use collapse, where visual tools like zoom-in cropping gradually disappear during RL fine-tuning even as task accuracy keeps improving. The key contribution is not just observing this collapse, but carefully diagnosing why forcing more tool use does not help. The authors show that rewarding tool frequency drives usage to near 100% but barely moves accuracy, because the model just keeps cropping the same regions without actually exploring more visual evidence. The fix they propose is simple and well justified， entropy regularization on the RL objective, while tool use still naturally fades away. The big concerns were scope (only one model and one tool zoom-in cropping), whether the gains really come from exploration diversity or just tool-use frequency, VLM-as-judge reliability, and early stopping. The authors handled it in the rebuttal, with new experiments with a 7-tool suite. So I recommended acceptance.